# External-Beam-Accelerated Partial-Breast Irradiation Reduces Organ-at-Risk Doses Compared to Whole-Breast Irradiation after Breast-Conserving Surgery

**DOI:** 10.3390/cancers15123128

**Published:** 2023-06-09

**Authors:** Oliver J. Ott, Wilhelm Stillkrieg, Ulrike Lambrecht, Claudia Schweizer, Allison Lamrani, Tim-Oliver Sauer, Vratislav Strnad, Christoph Bert, Carolin C. Hack, Matthias W. Beckmann, Rainer Fietkau

**Affiliations:** 1Department of Radiation Oncology, Universitätsklinikum Erlangen, 91054 Erlangen, Germany; willi.stillkrieg@uk-erlangen.de (W.S.); ulrike.lambrecht@uk-erlangen.de (U.L.); claudia.schweizer@uk-erlangen.de (C.S.); allison.lamrani@uk-erlangen.de (A.L.); tim-oliver.sauer@uk-erlangen.de (T.-O.S.); vratislav.strnad@uk-erlangen.de (V.S.); christoph.bert@uk-erlangen.de (C.B.); rainer.fietkau@uk-erlangen.de (R.F.); 2Comprehensive Cancer Center Erlangen-EMN, 91054 Erlangen, Germany; carolin.hack@uk-erlangen.de (C.C.H.); matthias.beckmann@uk-erlangen.de (M.W.B.); 3Department of Gynecology and Obstetrics, Universitätsklinikum Erlangen, 91054 Erlangen, Germany

**Keywords:** breast cancer, accelerated partial-breast irradiation, organs-at-risk, mean heart dose, mean lung dose

## Abstract

**Simple Summary:**

Compared to whole-breast irradiation, partial-breast irradiation uses smaller radiotherapy volumes and usually leads to lower radiation doses to the healthy tissues, such as the heart and lungs. For the first time, the present evaluation offers a complete analysis of the doses to on whole-breast irradiation. The dose reduction to the healthy organs was significant in favor of partial-breast irradiation. Therefore, partial-breast irradiation should be recommended to suitable patients to minimize the risk of secondary tumor induction and the incidence of consecutive major cardiac events.

**Abstract:**

In order to evaluate organ-at-risk (OAR) doses in external-beam-accelerated partial-breast irradiation (APBI) compared to standard whole-breast irradiation (WBI) after breast-conserving surgery. Between 2011 and 2021, 170 patients with early breast cancer received APBI within a prospective institutional single-arm trial. The prescribed dose to the planning treatment volume was 38 Gy in 10 fractions on 10 consecutive working days. OAR doses for the contralateral breast, the ipsilateral, contralateral, and whole lung, the whole heart, left ventricle (LV), and the left anterior descending coronary artery (LAD), and for the spinal cord and the skin were assessed and compared to a control group with real-world data from 116 patients who underwent WBI. The trial was registered at the German Clinical Trials Registry, DRKS-ID: DRKS00004417. Compared to WBI, APBI led to reduced OAR doses for the contralateral breast (0.4 ± 0.6 vs. 0.8 ± 0.9 Gy, *p* = 0.000), the ipsilateral (4.3 ± 1.4 vs. 9.2 ± 2.5 Gy, *p* = 0.000) and whole mean lung dose (2.5 ± 0.8 vs. 4.9 ± 1.5 Gy, *p* = 0.000), the mean heart dose (1.6 ± 1.6 vs. 1.7 ± 1.4 Gy, *p* = 0.007), the LV V23 (0.1 ± 0.4 vs. 1.4 ± 2.6%, *p* < 0.001), the mean LAD dose (2.5 ± 3.4 vs. 4.8 ± 5.5 Gy, *p* < 0.001), the maximum spinal cord dose (1.5 ± 1.1 vs. 4.5 ± 5.7 Gy, *p* = 0.016), and the maximum skin dose (39.6 ± 1.8 vs. 49.1 ± 5.8 Gy, *p* = 0.000). APBI should be recommended to suitable patients to minimize the risk of secondary tumor induction and the incidence of consecutive major cardiac events.

## 1. Introduction

Over the past decades, accelerated partial-breast irradiation (APBI) has become a standard treatment option for selected early-breast-cancer patients with a genuine low-risk of local recurrence after breast-conserving surgery [1]. Several well-powered randomized phase-III trials have compared different APBI techniques to standard whole-breast irradiation (WBI) and proved their safety and efficacy for a selected group of patients with more than 14,000 patients [2]. The most convincing data regarding local control and survival endpoints are available for APBI performed with external-beam radiation therapy or interstitial multi-catheter brachytherapy [3,4,5]. In general, the major advantages of APBI compared to standard WBI are the reduction in the overall treatment time and the comparably smaller treatment volumes and fewer early-onset side effects reported in many published trials [6,7]. Despite the existing body of evidence and the advantages of APBI, many radiation oncologists around the globe seem to be hesitant with its implementation into clinical routine. Reasons for this kind of reluctance among the radiation oncologists might be limited to familiarity regarding the potential benefits of APBI and concerns of decreasing the local control probability. However, in recent years, a growing awareness and concern about heart and lung irradiation doses has been recognized among the different medical professionals [8], as well as among the patients referred to radiation oncology departments for adjuvant breast irradiation. The question arises as to how long the patients are willing to accept a standard tangential WBI treatment, when a volume-limited APBI approach would be as justifiable in its safety and feasibility. Should the applied irradiation dose not be limited to be as low as reasonably achievable for the uninvolved healthy tissues (ALARA principle)? The present analysis was initiated to quantify the radiation doses for typical organs at risk (OAR) associated with linac-based external-beam APBI compared to standard WBI. To our knowledge, this is one of the first prospective reports with a representative and comprehensive dosimetric OAR documentation of all 170 study patients compared to representative real-world data from all 116 WBI patients who started adjuvant radiotherapy in 2020 at the same radiation oncology department.

## 2. Patients and Methods

Between 2011 and 2021, 170 patients with pT1-2pN0 breast cancer were enrolled in a prospective, mono-institutional external-beam APBI phase-II trial. All studies on humans described in the present manuscript were carried out with the approval of the local ethics committee and in accordance with national law and the Helsinki Declaration of 1975 in its current, revised form. Informed written consent was obtained from all patients. The APBI trial was registered at the German Clinical Trials Register, DRKS-ID: DRKS00004417. Detailed information on the study design, clinical endpoints, inclusion and exclusion criteria, and the treatment schedule has been reported previously [9,10].

Computed tomography (CT)-based irradiation planning and every radiotherapy fraction without exception were performed using a standardized breathing protocol (endexspiratory breath hold, EEBH) to decrease intrafractional motion at dedicated linear accelerators (Novalis classic, BrainLab AG, Munich, Germany; Vero SBRT System, BrainLab AG, Munich, Germany) combined with the ExacTrac^®^ X-ray patient position monitoring system (BrainLab AG, Munich, Germany), mandatorily using tumor bed clips in any fraction for exact positioning and minimization of interfractional positioning errors.

The planning target volume (PTV) consisted of the tumor bed with combined surgical and radio-oncological safety margins of 20 mm in each of the six directions. Target volume definition and delineation were performed in analogy to the recommendations for multicatheter interstitial brachytherapy [11], using a 15-mm expansion in each direction excluding lung tissue for the generation of the PTV from the clinical target volume (CTV). For dose-volume-histogram (DVH) evaluation, a modified PTVeval was restricted to the enclosed breast tissue by excluding the extracorporeal areas, skin, and chest wall muscles derived from the original PTV. The prescribed reference dose was 38 Gy in 10 fractions on consecutive working days with only one weekend in between. OAR contouring of the contralateral breast, the ipsilateral and contralateral lung volumes, the total lung volume, the whole heart, the left ventricle, the left anterior descending artery, the spinal cord, and the skin was consistently performed, and when additional auto-contouring software was used, checked and manually adapted exclusively by one experienced radiation oncologist (OJO) in all 170 patients. Standardized contouring of the heart substructures was performed according to the recommendations of Duane et al. [12].

Regarding the WBI group, all breast-cancer patients were included when starting adjuvant irradiation after breast-conserving surgery at our department in 2020. Patients with a palliative treatment approach because of metastatic disease or synchronous bilateral breast cancer were excluded from the analysis. Breast-conserving surgery, radiotherapy timing, and systemic therapies were performed by the same institutions as in the APBI group and followed similar departmental treatment guidelines based on the current version of the national treatment guidelines [13]. CT-based planning and radiotherapy delivery were routinely performed either with free breathing (FB), or deep inspiration breath hold (DIBH) techniques in the case of left-sided breast cancer without additional ipsilateral supraclavicular irradiation. Patient positioning was usually performed with a surface guided radiation therapy system (AlignRT^®^ version 5.1.2, Vision RT, London, United Kingdom) and 1 or 5 cone-beam CTs (CBCT) each week depending on individual errors judged by an experienced radiation oncologist after the first five fractions with CBCT-guided positioning [14]. The planning target volume (PTV) was delineated in accordance with the ESTRO guidelines on target volume delineation for elective radiation therapy of early-stage breast cancer [15]. The PTV encompassed the whole breast up to the inferior edge of the sterno-clavicular joint and included the adjacent thoracic wall. The ipsilateral supraclavicular lymph node area was included in cases of node-positive disease. The lymph nodes of the internal mammary chain were not irradiated electively. Patients received a total dose at the ICRU-50 reference point of either 50.4 Gy using mostly tangential fields (including intensity-modulated transmission fields) in 1.80 Gy daily fractions, five times a week, or a hypofractionated regime with 40.05 Gy (single dose 2.67 Gy, five times a week) or, for one patient, 26 Gy (single dose 5.20 Gy, five times a week). Boost irradiation to the tumor bed was performed with either photons, electrons, or interstitial brachytherapy; however, the additional dose contribution of the boost was not considered for this comparison.

OAR contouring for the WBI patients was performed in the exact same way as described above for the APBI patients, exclusively by the corresponding author. Typical OAR-dose values for the contralateral breast (dose maximum, mean dose), ipsilateral lung (D30, D2, D1: dose at a specific percentage of the corresponding volume; mean lung dose (MLD); V5, V10, V20, V30: volume fraction at a specific dose level), contralateral lung (D5, D2, D1, MLD, V5, V10), total lung volume (D2, D1, MLD, V5, V10, V20, V30), whole heart volume (D5, mean heart dose), left ventricle (D2, D1, mean dose, V5, V23), left anterior descending coronary artery (D2, D1, mean dose, V10, V20, V30, V40), spinal cord (maximum dose), and skin (D2, D1, maximum dose) were assessed in both groups.

Data management and statistics were carried out with IBM SPSS Statistics for MS Windows (SPSS Inc., Chicago, IL, USA), release 28. Dose distributions were analyzed with the Mann–Whitney U test. *p*-values < 0.05 were considered significant. The reported *p*-values are generally two-sided and considered to be explorative.

## 3. Results

A total of 286 patients were included in this dosimetric comparison. APBI was administered in 170/286 (59%) patients, WBI in 116/286 (41%). In 167/170 (98%) of the APBI patients, a complete dataset was available; among the remaining patients, two contributed OAR values in part (ipsilateral lung: D30; contralateral lung: D5; whole heart: D5, MHD; spinal cord: maximum dose; skin: maximum dose in one patient), and in the last patient, only the maximum doses for the spinal cord and the skin were accessible because of technical limitations. Regarding the WBI patients, the complete dosimetric OAR datasets were available in all cases.

The median age of the APBI patients was 62 years (range, 49–86), and for the WBI patients 56 years (range, 30–81), respectively (*p* < 0.001). This represents an expected difference, because patients < 50 years were excluded from participation in the APBI trial. Regarding the distribution between the left- and right-sided breast cancers, as well as the localization of the involved quadrant (lateral vs. medial/central), no differences were found between the groups. The size of the APBI-PTVeval was significantly smaller compared to the WBI-PTV (178 ± 86 vs. 836 ± 451 mL, *p* < 0.001). Further details on target volumes, breathing techniques, and radiotherapy dose regimes may be found in Table 1, and detailed information about the various OAR dose distributions is available in Table 2.

### 3.1. Contralateral Breast

The overall comparison between both groups revealed significantly lower mean and maximum dose values in the contralateral breast in favor of APBI. Regarding right- and left-sided breast cancer separately, the mean dose to the contralateral breast was ≤0.9 Gy in any subgroup and was significantly reduced with a factor of about two by the use of APBI (1.8–2.3). This effect appeared to be a little more pronounced for patients with right-sided breast cancer.

### 3.2. Lungs

Regarding the ipsilateral lung OAR doses, the results of all eight evaluated dose values were significantly lower with APBI compared to WBI (see Table 2). For example, the ipsilateral mean lung dose (MLD) with APBI was only 47% of that of WBI (4.3 ± 1.4 Gy vs. 9.2 ± 2.5 Gy, *p* = 0.000), and the V20 was about six times lower (3.0 ± 2.0% vs. 17.9 ± 4.9%, *p* = 0.000), respectively. The ipsilateral V20 reduction by the use of APBI compared to WBI is illustrated in Figure 1. The side-specific comparison showed similar effects among the patients with either left- or right-sided cancers. In addition, no differences were found for the six evaluated dose values of the contralateral lung. The contralateral MLD was ≤0.7 Gy in each scenario. The total lung OAR values were quite comparable to the results of the ipsilateral lung; every single comparison showed a favorable dose distribution for APBI patients, both in left- and right-sided breast cancers.

### 3.3. Heart

Depending on the subgroup, the mean total heart dose (MHD) ranged between 1.0 and 2.3 Gy. It was slightly lower with APBI for the whole collective, as well as for the right-sided breast-cancer subgroup, whereas the difference did not reach any significance for patients with left-sided breast cancer (see Table 2). The whole heart D5 revealed no statistical differences in all analyzed subgroups with APBI and WBI. The left ventricle V5 also showed no significant differences, whereas the left ventricle V23 was significantly lower in favor of APBI for both left- and right-sided breast cancers. The left ventricular peak values D1 and D2 illustrated a significant advantage for APBI for all patients and patients with left-sided breast cancer but no differences in right-sided cancers. The LAD mean dose was lower with APBI in any subgroup analysis; V10, V20, V30, V40, D1, and D2 were lower in the whole group and in the group of left-sided patients but not for the right-sided subgroup.

### 3.4. Spinal Cord and Skin

The average spinal cord maximum doses were lower for APBI compared to WBI patients (1.5 ± 1.1 Gy vs. 4.5 ± 5.7 Gy, *p* = 0.016) but were not significant among either the left- or right-sided subgroups. All skin-related peak dose values were significantly higher with WBI in all analyzed scenarios; for example, the D1 values were 35.7 ± 3.2 Gy for the APBI group and 47.2 ± 5.4 for the WBI group (*p* = 0.000). For further details, please refer to Table 2.

## 4. Discussion

### 4.1. Target Volumes

A major reason for decreased OAR doses in APBI compared to WBI is the smaller target volumes. However, for both techniques, representative data on PTV or CTV size are sparse, the reported volume definitions differ from study to study, and the resulting values are poorly standardized. In a recently published in silico planning trial, Duma et al. reported on 56 patients who were referred to adjuvant WBI radiotherapy after breast-conserving surgery [16]. The CTV was defined according to the European Society for Radiotherapy and Oncology (ESTRO) guidelines [15], and standardized PTVs were generated with an additional 1-cm safety margin. The resulting CTVs and PTVs were 632 mL (range, 109–1708 mL) and 1137 mL (range, 369–2500 mL), respectively. In the Danish randomized PBI phase III-trial, WBI CTV generation was also performed according to the ESTRO guidelines; however, this was achieved mostly by using an extension of 5 mm to create the PTV [17]. The median CTV was 663 mL (interquartile range, 443–994 mL). In the RAPID trial [18], the mean breast volume was 1490 ± 587 mL in the WBI group. In a smaller randomized APBI trial from Barcelona, the mean whole-breast reference volume was 1046 mL (range, 340–2577 mL) in the control WBI group. The whole-breast reference volume was defined as all tissue delimited by standard whole-breast tangent fields, excluding tissues deep in the chest wall, such as the lung, heart, pericardial fat, and liver [19]. In the present analysis, the mean WBI PTV size was 836 ± 451 mL, i.e., comparable or even slightly smaller than in the mentioned references.

The partial-breast irradiation PTV sizes published vary between 81 and 332 mL. In the randomized GEC-ESTRO APBI trial, interstitial multicatheter brachytherapy was used in the experimental arm and the volume enclosed by the reference isodoses was quite small at 81 mL (range, 7–275 mL) [4], whereas the biggest PTVs were reported in the RAPID trial with a mean V95 of 332 ± 153 mL for linac-based external-beam APBI [18]. In the Barcelona trial, the external-beam APBI PTV was 256 mL (range, 60–564 mL) [19]. The majority of the prospective series reported on PTVeval (usually defined as breast-tissue volume enclosed by the reference isodose with the exclusion of the skin and thoracic wall) values between 150 and 180 mL [20]. In the present analysis, the mean APBI PTVeval volume was 178 ± 86 mL, i.e., comparable to previously published data. In summary, comparing the available data, the target volumes were reduced by the use of partial-breast irradiation techniques with a factor of 4.0–5.6.

### 4.2. Contralateral Breast

A large epidemiological study with a total of 64,782 women who had received surgery following the first diagnosis of breast cancer revealed increased standardized incidence ratios for secondary breast cancer in patients with and without previous radiotherapy ≥ 5 years after the initial diagnosis [21]. The proportion of the second breast cancers which were contralateral was the same in the two cohorts (94%). Comparing both groups with and without radiotherapy, this effect was more pronounced in the irradiated group with a relative risk of 1.16 (95% CI 1.02–1.31). Another population-based study with 134,501 patients described an elevated long-term risk for contralateral breast cancer after radiotherapy for breast cancer with a 10- and 20-year actuarial rate of contralateral cancers of 6.1% and 12%, respectively, and concluded that unnecessary radiation exposure to the contralateral breast should be avoided for all patients with early-stage breast cancer [22].

The important Women’s Environmental, Cancer, and Radiation Epidemiology (WECARE) study analyzed the risk of second primary breast cancer in the contralateral breast following radiation therapy for first breast cancer [23]. This study compared 708 women with asynchronous bilateral breast cancer to 1399 women with unilateral breast cancer. The mean radiation dose to the specific quadrant of the contralateral breast tumor was 1.1 Gy. Women < 40 years of age who received >1.0 Gy to the specific quadrant of the contralateral breast had a 2.5-fold greater risk than unexposed women (RR = 2.5, 95% CI = 1.4–4.5). No excess risk was observed in women > 40 years of age.

Furthermore, various in silico WBI planning trials described contralateral breast mean doses for different scenarios (free breathing, breath hold, various field geometrics etc.) between 0.32 and 4.80 Gy [24,25,26,27,28,29,30,31,32,33,34]. In the present analysis, the mean dose to the contralateral breast was comparatively low at 0.80 Gy for all patients after WBI.

Robinson et al. determined the dose to the contralateral breast during APBI with breast high-dose-rate brachytherapy (SAVI Applicator, 10 × 340 cGy in five days) with thermoluminescent dosimeter (TLD) packets [35]. Measurements indicated an average total dose to the contralateral breast of 116 cGy for all 12 evaluated patients, and 70 cGy for the outer quadrants and 181 cGy for inner quadrant implants. Pignol et al. made a dosimetric in silico comparison of various breast irradiation techniques and found a slightly higher dose to the contralateral breast for interstitial high-dose-rate APBI (dose regime: 10 × 340 cGy, dose to contralateral breast: 230 mSv) than for external-beam 3D-conformal APBI (dose regime: 10 × 385 cGy, dose to contralateral breast: 140 mSv). For conventionally fractionated WBI up to 50 Gy, the dose to the contralateral breast was calculated with 1695 mSv for a wedge-based technique and 121 mSv for a IMRT-based technique [36]. In the present study, the mean dose to the contralateral breast was among the lower range of published values, for both the APBI and WBI patients, but the exposition was significantly lower with APBI (0.4 ± 0.6 Gy vs. 0.8 ± 0.9 Gy, *p* = 0.000), which is an interesting finding especially for the adjuvant radiotherapy of younger women.

### 4.3. Lungs

The risk of a second cancer such as lung cancer is increased after radiotherapy for breast cancer as well. A systematic review and meta-analysis of 762,468 patients revealed that previous radiotherapy due to breast cancer was significantly associated with an increased relative risk (RR) of secondary cancer of the lung ≥5 years after breast-cancer diagnosis (RR 1.39, 95% CI 1.28–1.51). The risk increased over time and was highest ≥ 15 years after breast-cancer diagnosis (RR 1.66; 95% CI 1.36–2.01) [37]. These findings emphasize the importance of reducing the OAR doses to a reasonable minimum without losing therapeutic efficacy (ALARA principle). 

A systematic review of lung-radiation doses from breast-cancer radiotherapy analyzed 471 WBI regimens from 32 countries. The average ipsilateral MLD was 9.0 Gy; for supine radiotherapy with no breathing adaption, it was 8.4 Gy; when the axilla/supraclavicular fossa was irradiated, it was 11.2 Gy; and with the addition of internal mammary chain irradiation, it was 14.0 Gy. Breathing adaptation reduced the ipsilateral MLD by 1 Gy, 2 Gy, and 3 Gy, respectively (*p* < 0.005). The average contralateral MLD was higher for IMRT (3.0 Gy) than for tangents (0.8 Gy) [38]. Another systematic literature review by the Early Breast Cancer Trialists’ Collaborative Group evaluated 647 radiotherapy dose regimens published between 2010 and 2015 and found an average dose of 5.7 Gy for the whole lung. Lung-cancer incidence ≥ 10 years after radiotherapy was increased compared to non-irradiated cases, yielding a rate ratio of 2.10 (95% CI, 1.48 to 2.98; *p* < 0.001) on the basis of 134 lung cancers, indicating a 0.11 (95% CI, 0.05 to 0.20) excess rate ratio per Gy whole-lung dose [39]. The overall results for the ipsilateral MLD were quite comparable to the WBI results in this study, and appeared to be even a little bit lower considering the high proportion of patients with an additional periclavicular target volume (47%, see Table 1). The average contralateral MLD was, with 60 cGy, lower among the WBI patients in the present analysis. The average doses to the whole lung were a little bit lower for WBI and only about half for APBI (see Table 2). In the review of Aznar et al., the ipsilateral MLD for partial-breast irradiation was 1.9 Gy (range, 0.3–5.0 Gy) [38]. The corresponding ipsilateral MLD results for the presented APBI trial were in the upper range with 4.3 ± 1.4 Gy, but the dose reduction to the ipsilateral lung was >50% compared to WBI (see Table 2). The ipsilateral V20 was reduced approximately six-fold by the use of external-beam APBI (see Figure 1). Additionally, the reduction in the OAR dose to the ipsilateral lung with APBI compared to WBI was confirmed (V10: 3.8 ± 3.9% vs. 6.5 ± 3.3%, *p* = 0.0001) in a randomized trial from Barcelona [19]. In analogy to the results of the presented data, an OAR difference for the contralateral lung was not described in the Spanish trial at all.

### 4.4. Heart

Today, it is widely accepted that radiotherapy for breast cancer may cause or aggravate severe cardiac disease [40,41]. In an influential population-based case-control study among 2168 women who underwent radiotherapy for breast cancer between 1958 and 2001 in Sweden and Denmark, major cardiac events increased linearly by 7.4% per Gy (95% confidence interval, 2.9 to 14.5; *p* < 0.001) of mean dose to the heart. The average mean dose to the whole heart was found to be 4.9 Gy [8]. The Early Breast Cancer Trialists’ Collaborative Group review estimated an average dose of 4.4 Gy for the whole heart [39]. In another systematic review of 398 analyzed regimens published between 2003 and 2013, the MHD was analyzed considering different clinical scenarios. The MHD for left- and right-sided breast cancer was 5.4 and 3.3 Gy; in regimens with and without the internal mammary chain, it was 4.2 and around 8 Gy; for intensity-modulated radiotherapy techniques, it was 5.6 Gy; and for tangential techniques with breathing control, it was 1.3 Gy. For partial-breast irradiation, it was quite low with 1.1 Gy. Interestingly, in a worldwide comparison of countries, the estimated average MHD in Germany showed the second highest value with 6.5 Gy, following Saudi Arabia with 7.9 Gy. The lowest average MHD was reported for the United Kingdom (1.6 Gy) [42]. 

Chiang et al. recently published a dosimetric in silico comparison of the OAR doses to the MHD and the LAD among 12 patients with interstitial brachytherapy APBI (26–28 Gy/4 fractions), external-beam APBI (38.5 Gy/10 fractions), and WBI plans (50 Gy/25 fractions) in left-sided early-breast-cancer patients. Compared to WBI, both APBI techniques showed a significant reduction for the MHD (3.2 ± 0.9 vs. 0.5 ± 0.2 and 1.1 ± 0.3 Gy) and mean LAD dose (3.3 ± 0.7 vs. 1.7 ± 1.0 and 0.5 ± 0.3 Gy). The OAR doses were equivalent in both APBI techniques [43].

Based on a literature review, an expert panel of the German Radio-Oncological Society (DEGRO) recommended the following constraints: MHD < 2.5 Gy; mean dose left ventricle <3 Gy; volume of left ventricle receiving ≥5 Gy < 17%; volume of left ventricle receiving ≥23 Gy < 5%; mean dose left descending artery < 10 Gy; volume of left descending artery receiving ≥30 Gy < 2%; volume of left descending artery receiving ≥40 Gy < 1% [44].

Our findings with an MHD of 1.6 ± 1.6 Gy for all APBI and 1.7 ± 1.4 Gy for all WBI cases corresponded with or were even lower than the results discussed from other cohorts. The MHD difference was small but significantly lower in favor of the APBI regimen (*p* = 0.007). The corresponding MHD values were slightly smaller for APBI in right-sided cancers (1.0 ± 1.1 Gy vs. 1.1 ± 1.1 Gy, *p* = 0.029) and were similar in left-sided ones (2.2 ± 1.8 Gy vs. 2.3 ± 1.4 Gy, *p* = n.s.). The DEGRO constraint values were calculated and are listed in Table 2; the recommended values were met for the majority of the patients and the results for the MHD, the mean dose of the left ventricle, the volume of left ventricle receiving ≥23 Gy, the mean dose of the LAD, the volume of the LAD receiving ≥30 Gy, and the volume of the LAD receiving ≥ 40 Gy were all significantly smaller with APBI compared to WBI (for further details, refer to Table 2).

### 4.5. Spinal Cord and Skin

The dose to the spinal cord is mainly dependent on whether the periclavicular region is included in the PTV or not. Regarding the whole cohort, our findings revealed a significantly lower maximum dose in favor of the APBI group (1.5 ± 1.1 vs. 4.5 ± 5.7 Gy, *p* = 0.016). Among the WBI patients, the maximum dose to the spinal cord was smaller compared to the APBI patients without irradiation of the periclavicular region (0.5 ± 0.3 Gy vs. 1.5 ± 1.1 Gy, *p* < 0.001) and clearly higher with this treatment (9.0 ± 5.5 Gy vs. 1.5 ± 1.1 Gy, *p* = 0.000). The maximum doses in the literature range between 0.18 Gy and 28.0 Gy depending on the particular technique used [45,46]. 

It has previously been reported in randomized trials that APBI techniques lead to reduced early skin toxicity in comparison to WBI regimes. For example, in the GEC-ESTRO trial, 1328 patients were either randomized to a conventionally fractionated WBI regime or APBI with interstitial multicatheter brachytherapy [4]. The incidence of grades 1–2 early skin side effects for WBI and APBI were 86 vs. 21% (*p* < 0.0001) [6]. Our current findings effectively support this advantage for APBI with significantly lower skin dose peak values of Dmax, D1, and D2 compared to WBI (see Table 2).

### 4.6. Summary

Our findings support the hypothesis that APBI compared to standard WBI regimens leads to significantly reduced relevant OAR doses, not only for the contralateral breast but also to the ipsilateral and whole lung, to the whole heart, as well as crucial cardiac substructures, to the spinal cord, and the skin. Patients selected for APBI treatments usually bear a low oncological risk of recurrence and death due to favorable biological and histopathological tumor features [47]. For these women, even a small increase in the risk for a second tumor in the contralateral breast or lungs or major cardiac events from radiation therapy may outweigh its benefits [42,48]. One limitation of our study is that it was based on a prospectively treated single-arm APBI study population compared to a retrospective real-world WBI collective with all the associated uncertainties of such an explorative data analysis.

## 5. Conclusions

In selected patients, APBI is proven to be equally effective regarding tumor control probability. Compared to WBI, smaller treatment volumes of APBI lead to lower OAR doses. In order to avoid unnecessary doses to healthy tissue, APBI should be recommended to suitable patients to consecutively minimize the risk of secondary tumor induction and the incidence of major cardiac events.

## Figures and Tables

**Figure 1 cancers-15-03128-f001:**
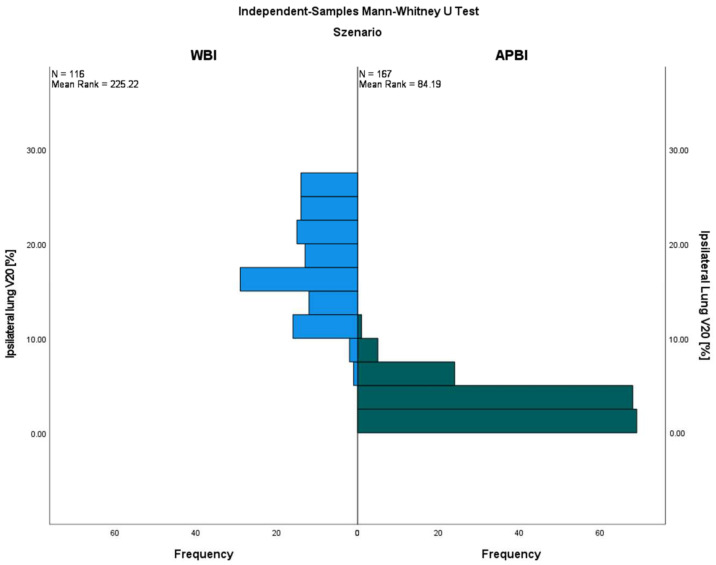
Ipsilateral lung V20 with external-beam APBI compared to the WBI control group.

**Table 1 cancers-15-03128-t001:** Patient and treatment characteristics.

	APBI (*n* = 170)	WBI (*n* = 116)	*p*-Value
Median age (range)	62 (48–86)	56 (30–81)	<0.001
Lateralization [*n*/*N* (%)]			
left-sided	84/170 (49)	61/116 (53)	n.s.
right-sided	86/170 (51)	55/116 (47)	
Tumor site [*n*/*N* (%)]			
lateral quadrants	100/170 (59)	66/116 (57)	n.s.
inner/central quadrants	70/170 (41)	49/116 (42)	
occult tumor site	-	1/116 (1)	
Radiotherapy technique [*n*/*N* (%)]			
partial breast	170/170 (100)	-	-
whole breast	-	62/116 (53)	
whole breast with PCF	-	54/116 (47)	
Breathing technique [*n*/*N* (%)]			
EEBH	169/170 (99)	-	-
FB	1/170 (1)	86/116 (74)	
DIBH	-	30/116 (26)	
Radiotherapy dose regimen [*n*/*N* (%)]			
10 × 3.80 Gy/1–2 weeks	170/170 (100)	-	-
28 × 1.80 Gy/5 weeks	-	60/116 (52)	
15 × 2.67 Gy/3 weeks	-	55/116 (47)	
5 × 5.20 Gy/1 week	-	1/116 (1)	
Mean PTVeval/PTV size [ml ± SD]	178 ± 86	836 ± 451	<0.001

APBI: external-beam accelerated partial-breast irradiation; WBI: whole-breast irradiation; Gy: Gray; PCF: periclavicular field; EEBH: endexspiratory breath hold; FB: free breathing; DIBH: deep inspiration breath hold; PTV: WBI planning treatment volume; PTVeval: APBI planning treatment volume restricted to the involved breast tissue; SD: standard deviation; n.s.: not significant.

**Table 2 cancers-15-03128-t002:** Organ-at-risk doses for APBI and WBI in left- and right-sided breast cancer after breast-conserving surgery.

	Left-Sided Breast Cancer			Right-Sided Breast Cancer			All Patients		
	APBI (*n* = 84)	WBI (*n* = 61)	*p*-Value	APBI (*n* = 86)	WBI (*n* = 55)	*p*-Value	APBI (*n* = 170)	WBI (*n* = 116)	*p*-Value
Contralateral breast									
▪ Dmax [Gy]	3.2 ± 5.7	8.2 ± 10.3	<0.001	3.3 ± 7.0	13.6 ± 13.5	<0.001	3.2 ± 6.4	10.8 ± 12.2	0.000
▪ Mean dose [Gy]	0.4 ± 0.6	0.7 ± 0.5	<0.001	0.4 ± 0.7	0.9 ± 1.2	<0.001	0.4 ± 0.6	0.8 ± 0.9	0.000
Ipsilateral lung									
▪ D30 [Gy]	4.9 ± 1.8	7.6 ± 4.0	<0.001	5.0 ± 1.9	7.7 ± 3.5	<0.001	5.0 ± 1.9	7.7 ± 3.7	<0.001
▪ V5 [%]	29.8 ± 12.6	35.1 ± 10.4	0.010	30.3 ± 13.5	36.1 ± 9.3	0.005	30.1 ± 13.1	35.6 ± 9.9	<0.001
▪ ▪ V10 [%]	10.8 ± 5.9	25.0 ± 7.1	0.000	10.1 ± 5.3	25.2 ± 6.5	0.000	10.5 ± 5.6	25.1 ± 6.8	0.000
▪ V20 [%]	3.1 ± 2.2	17.9 ± 5.2	0.000	2.9 ± 1.9	17.9 ± 4.5	0.000	3.0 ± 2.0	17.9 ± 4.9	0.000
▪ V30 [%]	1.1 ± 1.0	13.3 ± 4.5	0.000	1.0 ± 0.9	13.3 ± 3.9	0.000	1.0 ± 1.0	13.3 ± 4.2	0.000
▪ MLD [Gy]	4.3 ± 1.4	9.1 ± 2.7	0.000	4.3 ± 1.4	9.2 ± 2.3	0.000	4.3 ± 1.4	9.2 ± 2.5	0.000
▪ D1 [Gy]	28.4 ± 7.0	44.2 ± 5.6	0.000	28.6 ± 7.0	43.6 ± 5.1	0.000	28.5 ± 7.0	43.9 ± 5.3	0.000
▪ D2 [Gy]	24.4 ± 7.2	43.1 ± 5.5	0.000	24.3 ± 7.1	42.6 ± 4.9	0.000	24.3 ± 7.1	42.9 ± 5.2	0.000
Contralateral lung									
▪ D5 [Gy]	1.9 ± 1.6	1.4 ± 1.4	n.s.	1.4 ± 1.2	1.5 ± 1.6	n.s.	1.6 ± 1.4	1.4 ± 1.5	n.s.
▪ V5 [%]	0.6 ± 2.2	1.2 ± 6.1	n.s.	0.6 ± 3.4	1.6 ± 7.7	n.s.	0.6 ± 2.9	1.4 ± 6.9	n.s.
▪ V10 [%]	0.1 ± 0.3	0.1 ± 0.4	n.s.	0.1 ± 0.4	0.1 ± 0.8	n.s.	0.1 ± 0.4	0.1 ± 0.6	n.s.
▪ MLD [Gy]	0.7 ± 0.6	0.6 ± 0.7	n.s.	0.5 ± 0.6	0.6 ± 0.8	n.s.	0.6 ± 0.6	0.6 ± 0.7	n.s.
▪ D1 [Gy]	2.6 ± 2.4	3.3 ± 5.4	n.s.	2.1 ± 2.4	2.7 ± 2.9	n.s.	2.4 ± 2.4	3.0 ± 4.4	n.s.
▪ D2 [Gy]	2.3 ± 2.0	2.6 ± 5.1	n.s.	1.8 ± 1.9	2.2 ± 2.3	n.s.	2.1 ± 2.0	2.4 ± 4.0	n.s.
Total lung									
▪ V5 [%]	13.7 ± 5.6	16.7 ± 7.1	0.013	17.1 ± 7.8	20.5 ± 7.9	0.018	15.4 ± 7.0	18.5 ± 7.7	<0.001
▪ V10 [%]	5.0 ± 2.6	11.5 ± 3.3	0.000	5.6 ± 3.1	13.8 ± 4.1	0.000	5.3 ± 2.8	12.6 ± 3.9	0.000
▪ V20 [%]	1.5 ± 1.0	8.2 ± 2.3	0.000	1.6 ± 1.1	9.8 ± 2.6	0.000	1.6 ± 1.1	9.0 ± 2.6	0.000
▪ V30 [%]	0.5 ± 0.5	6.1 ± 2.1	0.000	0.6 ± 0.6	7.2 ± 2.3	0.000	0.5 ± 0.6	6.6 ± 2.2	0.000
▪ MLD [Gy]	2.3 ± 0.7	4.5 ± 1.4	0.000	2.6 ± 0.9	5.3 ± 1.5	0.000	2.5 ± 0.8	4.9 ± 1.5	0.000
▪ D1 [Gy]	24.7 ± 7.2	43.3 ± 5.5	0.000	26.1 ± 7.2	43.0 ± 5.0	0.000	25.4 ± 7.3	43.2 ± 5.2	0.000
▪ D2 [Gy]	19.0 ± 6.9	41.4 ± 5.4	0.000	19.8 ± 6.8	41.5 ± 5.0	0.000	19.4 ± 6.8	41.4 ± 5.2	0.000
Heart									
▪ Whole heart D5 [Gy]	5.9 ± 4.5	6.9 ± 6.8	n.s.	2.7 ± 2.6	2.4 ± 2.3	n.s.	4.3 ± 3.9	4.7 ± 5.6	n.s.
▪ MHD [Gy]	2.2 ± 1.8	2.3 ± 1.4	n.s.	1.0 ± 1.1	1.1 ± 1.1	0.029	1.6 ± 1.6	1.7 ± 1.4	0.007
▪ LV V5 [%]	18.6 ± 25.6	10.0 ± 12.2	n.s.	1.2 ± 8.3	1.6 ± 10.0	n.s.	9.9 ± 20.8	6.0 ± 11.9	n.s.
▪ LV V23 [%]	0.2 ± 0.5	2.6 ± 3.2	<0.001	-	-	-	0.1 ± 0.4	1.4 ± 2.6	<0.001
▪ LV Dmean [Gy]	2.7 ± 2.5	3.3 ± 1.9	0.007	0.6 ± 0.9	0.7 ± 0.9	<0.001	1.6 ± 2.1	2.0 ± 2.0	<0.001
▪ LV D1 [Gy]	9.3 ± 7.8	24.7 ± 16.8	<0.001	1.4 ± 1.8	1.1 ± 1.6	n.s.	5.3 ± 6.9	13.5 ± 17.0	0.002
▪ LV D2 [Gy]	8.2 ± 6.7	21.0 ± 16.7	<0.001	1.3 ± 1.7	1.1 ± 1.5	n.s.	4.7 ± 6.0	11.5 ± 15.7	0.003
▪ LAD V10 [%]	10.5 ± 20.9	19.1 ± 15.4	<0.001	0.8 ± 5.3	0.5 ± 2.8	n.s.	5.6 ± 15.9	10.3 ± 14.6	<0.001
▪ LAD V20 [%]	2.3 ± 7.9	13.3 ± 14.2	<0.001	-	-	-	1.1 ± 5.6	7.0 ± 12.2	<0.001
▪ LAD V30 [%]	0.7 ± 4.9	10.3 ± 12.4	<0.001	-	-	-	0.3 ± 3.5	5.4 ± 10.6	<0.001
▪ LAD V40 [%]	-	5.8 ± 11.2	<0.001	-	-	-	-	3.1 ± 8.6	<0.001
▪ LAD Dmean [Gy]	4.3 ± 4.0	8.3 ± 5.6	<0.001	0.8 ± 1.3	1.0 ± 1.3	<0.001	2.5 ± 3.4	4.8 ± 5.5	<0.001
▪ LAD D1 [Gy]	9.2 ± 8.9	29.9 ± 15.6	<0.001	1.8 ± 2.5	1.4 ± 2.1	n.s.	5.5 ± 7.5	16.4 ± 18.3	<0.001
▪ LAD D2 [Gy]	8.8 ± 8.5	28.5 ± 16.2	<0.001	1.7 ± 2.4	1.4 ± 2.1	n.s.	5.2 ± 7.2	15.6 ± 18.0	<0.001
Others									
▪ Spinal cord Dmax [Gy]	1.6 ± 1.2	4.3 ± 5.1	n.s.	1.3 ± 0.9	4.6 ± 6.3	n.s.	1.5 ± 1.1	4.5 ± 5.7	0.016
▪ Skin Dmax [Gy]	39.8 ± 2.2	49.3 ± 5.8	0.000	39.4 ± 1.3	48.8 ± 5.8	0.000	39.6 ± 1.8	49.1 ± 5.8	0.000
▪ Skin D1 [Gy]	35.6 ± 3.4	47.4 ± 5.4	0.000	35.8 ± 2.9	47.0 ± 5.4	0.000	35.7 ± 3.2	47.2 ± 5.4	0.000
▪ Skin D2 [Gy]	31.7 ± 5.2	46.8 ± 5.4	0.000	32.1 ± 4.8	46.3 ± 5.3	0.000	31.9 ± 5.0	46.6 ± 5.3	0.000

APBI: external-beam accelerated partial-breast irradiation; WBI: whole-breast irradiation; Gy: Gray; MLD: mean lung dose; MHD: mean heart dose; n.s.: not significant.

## Data Availability

The data presented in this study are available on request from the corresponding author. The data are not publicly available due to privacy restrictions and ethical issues.

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
