# Peer review of "External-Beam-Accelerated Partial-Breast Irradiation Reduces Organ-at-Risk Doses Compared to Whole-Breast Irradiation after Breast-Conserving Surgery"

_cancers, 2023, doi:10.3390/cancers15123128_

Round 1

Reviewer 1 Report

The authors report on a clinical trial aimed at reducing the radiation burden of breast cancer patients receiving radiotherapy. The study spanned 10 years and involved 286 patients split into two arms. The effects on organs such as lungs and heart were monitored. The results show a benefit for the patents in receiving focused doses rather than whole breast irradiation.

The manuscript is well written and organised. Table 2 is very large, if the authors can find a way to make it more concise then that would be helpful.

From the perspective of a non-clinical scientist working in anticancer drug development I find this study convincing and recommend for publication in the journal Cancers.

Small thing:

Line 56: “Radio-oncological departments around the globe seem to be hesitant” à departments are not people

Reviewer 2 Report

This study aimed to compare the doses of organs at risk (OAR) in patients who received external beam accelerated partial breast irradiation (APBI) versus those who underwent standard whole breast irradiation (WBI) after breast conserving surgery. The study included 170 patients with early breast cancer who received APBI, and their OAR doses were assessed and compared to a control group of 116 patients who underwent WBI. The results showed that APBI led to reduced OAR doses for the contralateral breast, the ipsilateral, contralateral and whole lung, the whole heart, left ventricle (LV) and the left anterior descending coronary artery (LAD), the spinal cord, and the skin, compared to WBI. Therefore, APBI should be recommended to suitable patients to minimize the risk of secondary tumor induction and the incidence of consecutive major cardiac events.

ALARA stands for "As Low As Reasonably Achievable," which is a principle used in radiation protection to minimize radiation exposure to patients, healthcare workers, and the public. In radiotherapy, the ALARA principle is applied to optimize the radiation dose delivered to the patient's tumor while minimizing the dose to surrounding healthy tissues. The goal is to achieve the most effective treatment with the least possible side effects.

To implement the ALARA principle in radiotherapy, common strategies such as IGRT have been used commonly. It uses imaging techniques such as CT scans, MRIs, and PET scans to precisely locate the tumor and surrounding healthy tissues before and during treatment. This allows for more accurate delivery of radiation to the tumor and less to surrounding healthy tissues.

1. secondary cancer projected risk models to evaluate the effect were not clarified.

2. Currently, APBI is used in selected low-risk patients with early-stage breast cancer. The NCCN Panel recommends APBI/PBI for any patient who is BRCA negative and meets the 2016 ASTRO criteria.     The 2016 ASTRO criteria define patients aged ≥50 years to be considered "suitable" for APBI/PBI if:  ◊  Invasive ductal carcinoma measuring ≤2 cm (pT1 disease) with negative margin widths of  ≥2 mm, no LVI, and ER-positive          or  ◊ Low/intermediate nuclear grade, screening-detected DCIS measuring size ≤2.5 cm with negative margin widths of ≥3 mm. 

3. This study compared a larger target to a small target to illustrate ALARA. The design was not sound.

Reviewer 3 Report

Title: External beam accelerated partial breast irradiation reduces  organ-at-risk doses compared to whole breast irradiation 3 after breast conserving surgery

The research work reported is interesting. There, however, are some concerns that the authors need to address (major revision):

1.    The abstract should be improved. You should state the main points of findings clearly. The current abstract includes many specific results, such as number and values. You should conclude your findings and potential implications in a broader view.

2.    While there are many interesting results in the main text, such findings are not adequately presented in the Abstract. I would suggest to add some key findings in the Abstract to attract readers.

3.    It has too brief in the 'Introduction' part (it should consist of about 450-600 words)

4.    The authors should highlight the manuscript's innovation and contribution

5.    The discussion part did not describe the future research plan, and did not describe where the next research of this article is directed.

6.    Kindly provide the limitations of proposed method and study.

Minor editing of English language required

Reviewer 4 Report

The study compared organs at risk (OAR) doses between accelerated partial breast irradiation (APBI) and standard whole breast irradiation (WBI). The results indicates that APBI should be recommended to suitable patients to minimize the risk of secondary tumor induction and the incidence of consecutive major cardiac events.

The reviewer thinks that the detailed study was performed, the study methods were precise and moderate, the results would be helpful to reduce the complications after radiation therapy. The reviewer thinks the article is worth being published, however, there are some concerns.

Line 157-158, in Results; The reviewer could not understand why the patients <50 years were excluded. All the exclusion or inclusion criteria should be clarified at the material and method section.

Reviewer 5 Report

The authors have evaluated the Organs at risk (OAR) doses in partial breast irradiation (APBI) Vs the whole breast irradiation (WBI) after breast conserving surgery using 170 APBI  and 116 WBI patients data. Overall it is an interesting study that support the hypothesis that APBI is better treatment option compared to WBI regimen. 

Comments:

1. What is the local reccurrence-free survival survival rate in APBI Vs WBI?

2. Is  overall survival is similar with APBI  compared to WBI?

3. what are the limitations of the current data shown? 

4. Some typos need to be corrected (ie., 14.000 patients in line 51 need to change to 14,000)

Round 2

Reviewer 3 Report

Overall, the authors have made substantial changes. The authors have responded to most of the comments. I will recommend to accept in the present form.

Minor editing of English language required